



# Site and Season Specific Calibrations Improve Low-cost Sensor Performance: Long-term Field Evaluation of PurpleAir Sensors in Urban and Rural India

Mark Joseph Campmier[1], Jonathan Gingrich[2], Saumya Singh[1], Nisar Baig[3], Shahzad Gani[4, 5], Adithi Upadhya[6], Pratyush Agrawal[7], Meenakshi Kushwaha[6], Harsh Raj Mishra[8], Ajay Pillarisetti[9], Sreekanth Vakacherla[7], Ravi Kant Pathak[8,10], and Joshua S Apte[1,9]

[1]Department of Civil and Environmental Engineering, University of California, Berkeley, CA, 94720, USA
[2]Department of Engineering, Dordt University, Sioux Center, IA, 51250, USA
[3]Department of Civil Engineering, Indian Institute of Technology Delhi, New Delhi 110016, India
[4]Centre for Atmospheric Sciences, Indian Institute of Technology Delhi, New Delhi 110016, India
[5]Institute for Atmospheric and Earth System Research/Physics, University of Helsinki, Helsinki, 00100, Finland
[6]ILK Labs, Bengaluru, India
[7]Center for Study of Science, Technology and Policy, Bengaluru 560094, India
[8]Indo Gangetic Plains-Centre for Air Research and Education, Hamirpur 210301, India
[9]School of Public Health, University of California, Berkeley, CA 94720, USA
[10]Department of Chemistry and Molecular Biology, University of Gothenburg, Gothenburg, Sweden

**Correspondence:** Joshua S. Apte (apte@berkeley.edu)

**Abstract.** We report on the long-term performance of a popular low-cost $PM_{2.5}$ sensor, the PurpleAir PA-II, at multiple sites in India, with the aim of identifying robust calibration protocols. We established 3 distinct sites in India (North India: Delhi, Hamirpur; South India: Bangalore), where we collocated PA-II with reference beta-attenuation monitors to characterize sensor performance and to model calibration relationships between PA-IIs and reference monitors for hourly data. Our sites remained

in operation across all major seasons of India. Without calibration, the PA-IIs had high precision (NRMSE among replicate sensors $\leq$ 15%) and tracked the overall seasonal and diurnal signals from the reference instruments well (Pearson's r $\geq$ 0.9) but were inaccurate (NRMSE $\geq$ 40%). We used a comprehensive feature selection process to create optimized site-specific calibrations. Relative to the uncalibrated data, parsimonious least-squares long-term calibration models improved PA-II performance at all sites (cross-validated NRMSE: 20-30%, $R^2$: 0.82-0.95), particularly by reducing seasonal and diurnal

biases. Because aerosol properties and meteorology vary regionally, the form of these long-term models differed by site. Likewise, using a moving-window calibration, we find a calibration scheme using seasonally specific information somewhat improves performance relative to a static long-term calibration model. In contrast, we demonstrate that a successful short-term calibration exercise for one season may not transfer reliably to other seasons. Overall, we demonstrate how the PA-II, when paired with a careful calibration scheme, can provide actionable information on $PM_{2.5}$ in India with only modest irreducible

uncertainty.



# 1 Introduction

Exposure to fine particulate matter, or PM$_{2.5}$ (particles with aerodynamic diameter $\leq$ 2.5 $\mu$m), is a leading cause of adverse health outcomes, including premature death (Lepeule et al., 2012; Vos et al., 2020). India experiences high mass concentrations in both its population dense megacities and its rural areas, resulting in the largest number of deaths (about 0.98 million annual

deaths, about 1.5 years reduction in life expectancy) attributable to ambient PM$_{2.5}$ worldwide (Apte et al., 2018; Pandey et al., 2021). In particular New Delhi, the surrounding Delhi National Capital Region, and the broader Indo-Gangetic Plain (IGP) of North India regularly experience hourly mass concentrations exceeding 1000 $\mu$g/m$^3$ (Gani et al., 2019) resulting in ill health effects even from short-term exposure (Gupta et al., 2021; Krishna et al., 2021). South India generally experiences lower PM$_{2.5}$ concentrations, but still has population-weighted annual mass concentrations that exceed World Health Organization

recommendations by a large margin (Apte and Pant, 2019). As relatively less polluted megacities in South India continue to rapidly grow, the challenge of ambient PM$_{2.5}$ will also increase (Guttikunda et al., 2019; Ramachandra et al., 2020).

Given the high exposure burden and complexity of PM$_{2.5}$ throughout India, there is need to increase understanding of the spatial-temporal patterns of air pollution. Traditional regulatory monitors are expensive to install and maintain, requiring specialized teams and consistent power to maintain networks (Brauer et al., 2019). As a result, there is a dearth of monitors in

India (Brauer et al., 2019; Martin et al., 2019). Although satellite remote sensing can fill in the spatial gap, it lacks high quality temporal coverage and relies on ground-based monitoring for calibration algorithms (Hammer et al., 2020), which can, as is the case in India, result in biased estimates of surface PM$_{2.5}$ (Dey et al., 2020).

Starting in around 2010, advancements in miniaturized electronics and laser technology have resulted in the growth of low-cost (< $500 USD) PM$_{2.5}$ sensor (LCS) technologies. These light-scattering monitors are popular within the research

community and among citizen scientists. The company PurpleAir (PA) has been especially successful in developing (1) a 200-280 USD LCS that utilizes a commercially available, light-scattering sensor developed by Plantower (PMS5003) and (2) a platform for individuals and organizations to share data from indoor and outdoor PurpleAir LCS.

Light-scattering LCS require extensive data quality control and careful selection of calibration models to offer measurements comparable to reference quality instruments (Hagan and Kroll, 2020; Hagler et al., 2018). Optical sensors mischaracterize mass

from aerosol scattering properties, as PM$_{2.5}$ is a mixture of particle sizes and chemical compositions (Hagan and Kroll, 2020; Levy Zamora et al., 2019; Zou et al., 2021). The roles of relative humidity, mass concentration range, sensor aging, and diverse source profiles have been extensively studied in laboratories and field conditions in the US, Australia, and Europe. Lab studies report the Plantower sensors fail to characterize fine particles above 0.8 microns (Kuula et al., 2020), deteriorate under extreme mass concentrations (Mehadi et al., 2020; Tryner et al., 2020), and are vulnerable to overestimation at RH greater than 60%

(Jayaratne et al., 2018).

Field studies in low to moderate pollution environments show PA units can be calibrated to reference instruments using simple empirical regression techniques with environmental variables (Barkjohn et al., 2021; Malings et al., 2019; Zheng et al., 2018). Models are often specific to a season and location, however, Barkjohn et al. (2021) demonstrated that a continental US calibration equation could be effectively deployed for daily data.



Recently there is increased interest in understanding LCS performance in the Global South to fill major monitoring gaps (Bai et al., 2020; Jha et al., 2021; Malyan et al., 2023; McFarlane et al., 2021; Puttaswamy et al., 2022; Sreekanth et al., 2022; Zheng et al., 2018, 2019). In North India, Zheng et al. (2018) deployed Plantower models in Kanpur, Uttar Pradesh, for 90 days and found multilinear regression improved Plantower performance, albeit with significant error for hourly data. In South India, Puttaswamy et al. (2022) calibrated Plantower units for 68-days in Chennai and found a multilinear regression approach

reduced uncertainty to within 15% and 18% for $PM_{2.5}$ and $PM_{10}$ respectively. LCS studies in India report the importance of climate and emissions variability on aerosol characteristics and advise future deployments to test calibration algorithms across longer timelines (Malyan et al., 2023; Puttaswamy et al., 2022; Sreekanth et al., 2022; Zheng et al., 2018, 2019).

In this study, we deployed and evaluated PurpleAir PA-II sensors in Delhi, Hamirpur, and Bangalore by collocating with regulatory grade instruments for 335, 154, and 312 days respectively. We built hourly local calibration models using multilin-

ear regression. With proper data quality constraints, a relatively simple calibration model can produce high accuracy and low biased data. Despite this success, model performance degrades when attempting to transfer a model trained in each environment to data collected in a dissimilar environment. We found more pronounced reduction in performance when attempting to transfer a model trained in one season to another season, as aerosol characteristics can shift rapidly even at the same site. Our work demonstrate LCS are a viable option for measuring spatial-temporal trends throughout India, but calibration models are

vulnerable to the local and seasonal effects on aerosol properties.

## 2 Methods

### 2.1 Low-cost Sensors

The sensor used in this study was the PurpleAir PA-II. The PA-II is marketed as PurpleAir's outdoor aerosol monitor, composed of a weatherproof plastic shell containing two Plantower PMS5003 sensors (labeled as "A" and "B" channels), an Adafruit

BME280 atmospheric sensor (temperature, RH, and pressure), and a wireless transmitter module to upload data via WiFi. The PMS5003 reports particulate matter (PM) mass concentrations ($\mu g/m^3$) of all particles with aerodynamic diameter smaller than 1 $\mu m$, 2.5 $\mu m$, and 10 $\mu m$, as well as particle number concentrations ($dl^{-1}$) of all particles larger than 0.3 $\mu m$, 0.5 $\mu m$, 1.0 $\mu m$, 2.5 $\mu m$, 5 $\mu m$, and 10 $\mu m$ (Zhou and Zheng, 2016)).

PA-IIs reports mass concentrations in two forms: CF1 and ATM – the "uncorrected" data and atmospheric corrected data,

respectively. Other recent studies have shown a method (denoted ALT) using $PM_{2.5}$ derived from the number concentration data to be comparable to and more transparent than the ATM data (Wallace et al., 2021, 2020). Additional details about the PA-II channels are presented in SI Fig. S1. In this analysis we use all three in our comparisons with and calibration to regulatory quality data.





## 2.2 Regulatory-grade Monitors

We compared our PurpleAir measurements against US EPA Federal Equivalent Method (FEM) certified continuous monitors. Our selected FEMs are MetOne BAM models 1020 and 1022, widely used devices (Hall and Gilliam, 2016) that use the beta wave attenuation technique to determine particle mass based on a sample deposited on a filter tape. FEM certification applies to 24-hour averaged data, while can the BAMs provide measurements at hourly or higher time resolution. We used the 1-hour block as our highest level of temporal resolution, similar to other LCS calibration studies using beta attenuation reference
monitors in the US and India (Johnson et al., 2018; Magi et al., 2020; Sreekanth et al., 2022; Zheng et al., 2018).

At the Delhi site, we used the BAM-1020; data from this monitor are public and maintained by the US State Department's AirNow service (San Martini et al., 2015). The Hamirpur and Bangalore sites utilized BAM-1022s managed in collaboration with field teams from the Indo-Gangetic Plains Center for Air Research and Education (IGP-CARE) and the Center for Study of Science, Technology, and Policy (CSTEP), who manually retrieved data at regular intervals. Staff at each site followed the
manufacturer's recommended operation and maintenance, which resulted in downtime for each dataset.

## 2.3 Deployment Sites

Three separate long-term measurement efforts were conducted to evaluate the PA-II performance under different meteorological and aerosol composition regimes. Each campaign was scheduled to last approximately one year, enabling comparison of a range of mass loadings and the effect of season. We use the Indian Meteorological Department's (IMD) definition of four
seasons: Winter (January, February), Pre-Monsoon (March, April, May), Monsoon (June, July, August, September), and Post-Monsoon (October, November, December). (Dubey et al., 2021). A reference map of the collocation sites is presented in SI Fig. S2.

### 2.3.1 US Embassy, New Delhi, National Capital Territory of Delhi, India

The Indian National Capital Region (NCR), including the capital city of New Delhi (elevation about 230 m), is the second
largest megacity in the world with a metro-area population around 28.5 million people. It has also been called the most polluted megacity in the world, experiencing annual average $PM_{2.5}$ concentrations exceeding 120 $\mu g/m^3$, with peak daily (hourly) in excess of 500 $\mu g/m^3$ (1500 $\mu g/m^3$) during post-monsoon and winter pollution episodes (SI Fig. S3, Gani et al. (2019)). The NCR along with the rest of the IGP experiences dynamic meteorology with cold wet winters, warm drier post-monsoons and pre-monsoons, and hot wet monsoons (SI Fig. S4).
Our measurement site was the US Embassy (28.5975 °N, 77.1878 °E) in the Chanakyapuri neighborhood of central New Delhi. The embassy is located within the city's spacious diplomatic enclave, which has abundant greenspace, relatively low traffic flows and minimal local industrial emissions. We collocated 2 PA-II units with the embassy BAM from 2018 July - 2020 April. During the course of our campaign, Delhi experienced extreme $PM_{2.5}$ concentrations during the post-monsoon agricultural burning seasons and characteristic winter inversion layers, with a relatively low-pollution monsoon season consistent with
expected seasonal trends (Guttikunda and Gurjar, 2012).



### 2.3.2 IGP-CARE, Hamirpur, Uttar Pradesh, India

We established a rural $PM_{2.5}$ monitoring site in Hamirpur district, located within the IGP in India's most populous state, Uttar Pradesh (UP). Our monitoring site was established in partnership with the Indo-Gangetic Plains Center for Atmospheric Research and Education (IGP-CARE). This remote solar-powered rural monitoring site is situated on a rooftop (20 m above

ground level) of a solitary building ((25.9552 °N, 80.1522 °E)) located about 800 m outside Ruri Para village in Hamirpur district, Uttar Pradesh. The immediate surroundings within 500 m of the site are a mixture of agricultural fields, ravines, and scrubland forest. The closest major town, Hamirpur (population about 35,000) is approximately 30 km away from the site, and the closest large city, Kanpur (population about 3 million) is 80 km away. Meteorological patterns are similar to Delhi (SI Fig. S5). We collocated three PA-II sensors with a BAM-1022 on the IGP-CARE rooftop beginning in January 2020. Here, we

report on data for the year from 2020 January to 2021 January.

Although campaign-median $PM_{2.5}$ concentrations at the IGP-CARE site (Table 1) are high in global context, this site's remote location outside of both cities and villages means that concentrations do not reach the same peaks as in Delhi. However, there are still many local sources of aerosol air pollution in rural North India such as biomass burning for cooking and heating (Rooney et al., 2019). The Hamirpur dataset is additionally differentiated from the Delhi dataset in that most of the data was

collected during the first year of the COVID-19 pandemic, which was observed to change patterns of emissions throughout the IGP (Patel et al., 2021; Singh et al., 2020).

### 2.3.3 CSTEP, Bangalore, Karnataka, India

Bangalore, in South India, is the third largest city in India, with a population of 8.4 million, and the capital of Karnataka. South India experiences different meteorological conditions and considerably lower air pollution burdens than North India

(Apte and Pant, 2019; Dubey et al., 2021) (SI Fig. S6, SI Fig. S7). Although continuous $PM_{2.5}$ regulatory monitors are sparse in Bangalore, the current network estimates a citywide annual average of 30 $\mu g/m^3$. While this value is low in comparison to cities in the IGP, it exceeds the WHO guideline value of 5 $\mu g/m^3$ and hourly winter concentrations often exceed 50 $\mu g/m^3$, with emissions dominated by traffic and dust resuspension (Guttikunda et al., 2019). In Bangalore, winters are milder, and the climate is more consistent year-round than in the IGP (SI Fig. S6). The winter, and pre-monsoon seasons are distinguished

from the monsoon and post-monsoon seasons primarily by RH and precipitation. Monsoon and post-monsoon are cloudy and rainy, with RH typically exceeding 70% all day, and can remain above 90% before sunrise. Winter, and pre-monsoon RH are more moderate with hourly averages fluctuating between 40-80%.

Our collocation site was the CSTEP office in northern Bangalore. CSTEP maintained a BAM-1022 on the rooftop of their 3-story office building (13.0485 °N, 77.5795 °E). Although the site is located near a highway (Outer Ring Road), the annual

diurnal patterns matched the regional signature from the average of the regulatory monitors. Furthermore, area surrounding CSTEP is mostly office buildings, with some residential housing. There are no large industrial sites or obvious large point sources in the neighborhood, other than occasional small solid waste fires. It is likely the CSTEP BAM is thus mostly influenced by urban background and regional aerosol conditions. We set up 2 PA-II sensors from 2019 June – 2020 July, during which



Bangalore experienced hourly spikes above 100 $\mu$g/m$^3$ during the festival of Diwali and dynamic changes in traffic patterns
due to the COVID-19 pandemic and lockdowns.

## 2.4   Quality Assurance

### 2.4.1   PurpleAir PA-II PM$_{2.5}$

Many light-scattering PM$_{2.5}$ sensors, including the PA-II, can report unrealistic measurements, lack accuracy (especially at
high mass loadings), and are only recommended for operation within a specific range. To minimize these effects, we removed
unreasonably large and small points, block-averaged each individual Plantower unit, averaged across all units for a given site,
removed imprecise points, and calibrated the resulting clean dataset. We conducted QA procedures separately for each sensor
correction factor (CF1, ATM, ALT).

We removed all raw PM$_{2.5}$ data points outside of the range 5 – 500 $\mu$g/m$^3$ (Kelly et al., 2017; Magi et al., 2020; Zhou and
Zheng, 2016). Analyses of PurpleAir data typically report the percent error between channels A and B for a given unit to remove
imprecise points, treating them as statistically paired measurements and all other nodes as statistically independent (Barkjohn
et al., 2021). However, at our collocation sites, there was always more than one PA-II, so we treated all Plantower sensors as
replicate measurements and averaged them together as a single data point. For instance, if we had three PA-IIs at a site, we
averaged the six values together – two from each unit – to estimate a single datapoint. We established 80% completeness criteria
for each hourly block average, and at least 2 valid Plantower block averages for the resulting site PA data point. Imprecise site
points were removed using the coefficient of variation (CV), the quotient of the mean and standard deviation of the sensors for
a given 2-min raw sample. CV values greater than 0.2 were removed, broadly consistent with approaches used by other studies
(Badura et al., 2018; Crilley et al., 2018).

### 2.4.2   PurpleAir PA-II Temperature and Relative Humidity

The BME280 is considered a reliable and accurate low-cost environmental sensor (Araújo et al., 2020). There are occasional
sensor miscommunications with the microprocessor, leading to unrealistic values, which we filtered out by restricting RH to 0-
100% and temperature to -10-50°C. We computed dew point temperature from the measured temperature and RH like Malings
et al. (2019).

### 2.4.3   MetOne BAM-1020, BAM-1022

The BAM instrument flags low quality data with a specific code to (1) potentially remove them from analyses and (2) to
diagnose underlying issues, which can include power loss and pump errors. The BAM-1020 and BAM-1022's default concen-
tration range is 3 - 1000 $\mu$g/m$^3$. Unlike the PA-II, the hourly LOD of the BAM-1022 and BAM-1020 is well constrained to 2.4
$\mu$g/m$^3$ (Magi et al., 2020), considerably below typical concentrations in our dataset. Like other linear regression studies using
MetOne BAMs models and Plantower nephelometers, we utilized an ordinary least squares (OLS) approach (Barkjohn et al.,
2021; Malings et al., 2019; McFarlane et al., 2021; Mehadi et al., 2020; Wallace et al., 2021; Zheng et al., 2018).





## 2.5 Calibration Regression

Since nephelometers and other optical based sensors are known to provide biased measurements of $PM_{2.5}$ measurements relative to reference grade instruments, in large part due to hygroscopic growth, calibration procedures attempt to account for bias due to RH. One approach is to leverage the environmental data (RH, temperature, etc.) from LCS nodes to develop the best fitting model without imposing any a priori assumptions about aerosol growth or chemistry. We label this approach as "data-driven." From decades of work with optical instruments, corrections have been developed assigning exponential growth terms as a function of RH and known $PM_{2.5}$ chemical characteristics. In our work, we label this approach as "theory-driven" since it attempts to fuse the best fitting function form from theory with the best fitting regression coefficients. Although the theory-driven model should produce the most transferable models since theory should apply in all environments, the underlying data processing of the Plantower - a truncated nephelometer (Ouimette et al., 2021) - may result in bias structure better explained by a linear RH correction than an exponential correction.

### 2.5.1 Data-Driven Model Selection

To ensure our work is easily reproducible within India, we relied only upon variables reported or calculable by the PA-II as independent variables: PM, RH, temperature, and dew point. For our PA-II $PM_{2.5}$ variable we evaluated CF1, ATM, and ALT values. We evaluated all regression models using OLS with the BAM $PM_{2.5}$ as the dependent variable and our candidate parameters as independent variables. To iterate across all possible arrangements of predictors - including additive terms, interaction terms, as well as polynomial terms up to order 3 – we implemented Sequential Feature Selection (SFS) using the Python package scikit-learn 0.24.2. SFS produces the most relevant features for a given a priori feature number by iteratively removing features and observing the impact on a performance metric, Bayesian Information Criterion (BIC) in our case. A user can then iterate across the number of features and compare the best performing models for each number of explanatory variables.

### 2.5.2 Theory-Driven Model Selection

From $\kappa$-Köhler theory, we expect wet PM scattering to increase exponentially with increasing RH, resulting in strongly non-linear dynamics. Therefore, we applied a calibration function relying on empirically fitted coefficients from the training data, with a non-linear RH term to capture expected trends from theory. Studies have attempted applying a non-linear RH term for light scattering LCS, with results similar to or less accurate than an additive term (Chakrabarti et al., 2004; Malings et al., 2019; Tryner et al., 2020; Zheng et al., 2018). Given the difference in emission sources, size distribution, mass loadings, and meteorology, we decided to include a non-linear RH term using the following form, Eq. (1).

$$C = \frac{\alpha \times P}{1 + \beta \frac{RH^2}{1-RH}} \tag{1}$$

Where $\alpha$ and $\beta$ represent regression coefficients to be fitted via Non-linear Least Squares, P is the PurpleAir signal (ATM, CF1, or ALT), RH is the unitless relative humidity scaled from 0 to 1, and C represents the corrected $PM_{2.5}$.





### 2.5.3 Cross-validation

To evaluate our calibration models, we sought to design an appropriate cross-validation scheme that would permit a balanced evaluation of model performance among all seasons. A simple test-train split would likely overrepresent seasons with more measurements. We thus performed a stratified k-fold cross-validation, in which each fold contains equal representation from each of the 4 seasons; we evaluated each model by leaving one-fold out in subsequent iterations.

### 2.5.4 Temporal Sensitivity

As a point of contrast with the seasonally balanced calibration described above, we performed to a data experiment to investigate the temporal stability of a hypothetical shorter-term calibration. This exercise was motivated by the common practice in many LCS deployments, of performing a short-term initial calibration, then deploying sensors in the field, and if the LCS are available, perform another short post-study collocation. To explore the potential bias from extrapolating a short-term calibration to a longer period, we fitted 4-week rolling OLS (ROLS) models with the features selected via SFS and compared the performance against all other 2-week periods during our yearlong data collection to understand the implications of short-term calibration for other studies.

### 2.5.5 Performance Metrics

As a guiding principle, we selected for presentation those models which balanced parsimony with low error, low bias, and strong temporal consistency. We selected analytical methods and performance metrics to optimize these parameters and have designated these best performing models as "robust." Given the high concentrations and high variability within and between sites, we report the normalized RMSE (NRMSE), allowing comparison of model performance across sites and time periods. Additionally, we used the coefficient of determination ($R^2$) to evaluate model accuracy. For multivariate regression models we used the adjusted $R^2$ metric to account for spurious correlations with increasing numbers of independent variables. To penalize overfit and minimize the number of parameters, we used the Bayesian Information Criterion (BIC) when selecting between models during the SFS process. Finally, we assessed the mean bias error (MBE) as well as normalized mean bias error (NMBE) to characterize the average direction of error.

## 3 Results and Discussion

### 3.1 Reference Instrument Data Summary and Quality Assurance

BAM and PA measurement summary statistics are summarized in Table 1 for each site, with time series plots in SI Fig. S8-S10. Overall, BAM monitors used at each site provided consistent performance despite challenging deployment circumstances due to intermittent power loss; extreme weather, including heavy rains; and a relatively broad range of mass concentrations.

The US State Department monitor in Delhi employs the US EPA's data reduction process, resulting in loss of about 3% of data points, with a continuous gap from 2019 February 10 to 2019 March 18. For context, we compared this site's time series





with 39 other sites in Delhi's regulatory network and found an $R^2$ of 0.86 and a mean bias of -8.41 $\mu$g/m³, likely resulting from this monitor's location in one of the city's cleanest neighborhoods. The diurnal plot for the Delhi BAM in Fig. 1 reflects the roles of time-varying emissions and boundary layer dynamics with peaks during the morning traffic rush hour (7-10 AM), and extremes in the winter exceeding an average of 200 $\mu$g/m³ during the night and early morning. During the monsoon, we observed a relatively low daily dynamic range of 35–50 $\mu$g/m³.

At both the IGP-CARE site in Hamirpur and the CSTEP site in Bangalore, we used the manufacturer's specified data flags to perform quality assurance, resulting in 6% and 11% data loss for the IGP-CARE site and CSTEP site BAMs, respectively. Unlike Delhi, the Bangalore network is sparse (n = 40 in Delhi versus n = 8 in Bangalore), withe relatively low data completeness from the official monitors. Diurnal plots in Fig. 1 show a morning peak, with maximum values typically at 8-9 AM for the collocation site BAM.

The closest regulatory monitor to the Hamirpur site is in Kanpur, more than 50 km away, too far for meaningful comparisons of local conditions. Figure 1 shows similar trends to the US Embassy site in Delhi, with a morning peak between 7 and 9 in the morning, extreme mass concentrations throughout the winter, and low dynamic range during the monsoon. There are no long continuous gaps from this monitor; however, power outages were more frequent in Hamirpur than the other two sites since it is a rural site, leading to significant data loss – about 14% of the total campaign hours, concentrated in the Pre-Monsoon.

## 3.2 PA-II Quality Assurance

We evaluated the unit-to-unit precision of the PA-II sensors by comparing the individual channels of all co-located Plantower sensors at each site. Because each PA-II contains two Plantower sensors, there were always a minimum of four Plantower sensors operating at each monitoring site. The PA-II PM$_{2.5}$ channels were highly precise, with strong correlation ($R^2 \geq 0.9$) both within nodes and between nodes across the mass concentration distribution, consistent with existing literature (Kelly et al.,

2017; Levy Zamora et al., 2019; Sahu et al., 2020). Bland-Altman plots indicate high precision across all sites and units, with mean differences centered near 0 $\mu$g/m³, and most hourly points within $\leq$ 20% (SI Fig. S11-S13). The between-Plantower $R^2$ for the CF1 data across all collocated PA-II sensors was between 0.94-0.99 for the Delhi site, 0.92-0.99 for the Bangalore site, and 0.95-0.99 for the Hamirpur site (SI Fig. S14). Disagreement was more pronounced at high concentrations (>100 $\mu$g/m³) at which $R^2$ ranges at each site dropped to 0.90 – 0.95, 0.83 – 0.88, and 0.92 – 0.94 for Delhi, Hamirpur and Bangalore

respectively. Similar intra-sensor correlations were found for the ATM and ALT data. Given the consistent between-sensor hourly precision across sites (NRMSE $\leq$ 10%), we can confidently state we expect random error of at most 10%.

   Applying the detection limit thresholds removed 1% of the total Delhi dataset, and <1% from the Hamirpur and Bangalore datasets. The CV test removed 15%, 15%, and 14% from the Delhi, Hamirpur, and Bangalore datasets averaged across CFs, with the differences across sites likely due to the higher average mass concentrations in the IGP. RH and temperature

microcontroller errors were limited to about 4% of the total data in Delhi and Hamirpur and <1% in Bangalore.

   After removing the filtered data points, accounting for power losses, and applying the completeness criteria for 1-hr block averages, the site averaged PA data resulted in average coverage of 47% (N=9260 hours), 63% (N=5958 hours), 86% (N=8567 hours) for Delhi, Hamirpur, and Bangalore respectively across CFs. Finally, the reference dataset was synchronized with the





PA dataset and the combined dataset coverage is 38% (N=7504), 39% (N=3744), 75% (N=7473) for Delhi, Hamirpur, and
Bangalore respectively.

### 3.3    PurpleAir Data Summary

Across sites, the PA-II captured diurnal and seasonal trends with similar results to collocated BAMs, as evident in Fig. 1 and SI
Fig. S15. However, inconsistent biases among season and location were also observed for all three $PM_{2.5}$ channels (CF1, ATM,
ALT), resulting in poor accuracy for the uncalibrated dataset. Although poor accuracy is unsurprising, our findings highlight
the importance of dynamic emissions and meteorology across the Indian subcontinent as well as field performance at extreme
mass concentrations.

In Delhi, the PA data (CF1) correctly identified winter and post-monsoon as the most polluted seasons, with strong diurnal
range peaking 8-9 AM (Fig. 1). The PA also well characterized the Delhi monsoon, with low diurnal range and a daily average
less than 60 $\mu$g/m$^3$. The uncalibrated LCS overestimates concentrations during the extremely polluted and humid post-monsoon
and winter, with a strong underestimate during the dry and hot pre-monsoon. The PA units at Hamirpur follows a similar
trend. Although both IGP sites feature relatively low bias in the pre-monsoon period, they frequently underestimate mass
concentrations in this season, perhaps due to the influence of coarse mineral dust (particle diameter > 1 $\mu$m), as observed
elsewhere in field and lab evaluations (Kuula et al., 2020; Levy Zamora et al., 2019; Sahu et al., 2020; Sayahi et al., 2019).

Since Bangalore's meteorology exhibits comparatively low seasonality, and emissions are more strongly influenced by mo-
bile sources rather than the more complex mixture in the IGP, LCS performance is different than in North India. During the
day (9 AM - 7 PM), accuracy is biased by more than +25% during the winter, pre-monsoon, and post-monsoon, with systemi-
cally lower bias including underestimates in the less polluted monsoon season (Fig. 1). Accuracy is lower during higher mass
loadings at night and during early morning hours, with strong overestimates across seasons, peaking during the most polluted
hour (7 - 8 AM).

### 3.4    Model Selection

#### 3.4.1    Data-Driven Model Fitting

The SFS procedure results are summarized in Table 2 (with extended results in SI Tables S1-S3), where the four most relevant
parameters are listed in order of decreasing importance for each CF and site. Across sites, $R^2$ stabilized at 2 parameters (about
0.8 for Delhi, and about 0.9 for Hamirpur and Bangalore). For all sites, sensor estimated $PM_{2.5}$ was generally selected as the
single most relevant parameter for predicting concentrations measured by BAM followed by a variation of RH (i.e., RH2,
RH3). The form of the most robust Bangalore model is different from the IGP sites with ALT $PM_{2.5}$ (rather than CF1 $PM_{2.5}$)
selected as the most predictive $PM_{2.5}$ data stream. Furthermore, the Bangalore dataset ranked temperature and dew point as
more relevant than the Hamirpur and Delhi datasets. Constraining Bangalore to the same top parameters as the North India
sites (CF1 $PM_{2.5}$ and RH) reveals only marginal differences ($\Delta$NRMSE $\approx$ 2%) in performance from the most robust model





selected by SFS (ALT $PM_{2.5}$ and $RH^3$). As such we choose to standardize our calibration across all sites with only CF1 $PM_{2.5}$ and RH as relevant parameters.

Regression coefficients of CF1 $PM_{2.5}$ data were positive values less than 1, indicating the CF1 data generally overestimate, but are positively correlated with reference monitors. RH term coefficients at the IGP sites are negative, indicating increasing RH should negatively weigh the PA reading, consistent with the expected artifacts of hygroscopic growth in the atmosphere.

The Bangalore dataset similarly assigns RH terms a negative weight. Temperature and dewpoint terms receive both negative and positive coefficients across orders of magnitude, and it is not determinable if the models are deriving a spurious correlation or detecting underlying aerosol or instrument properties.

### 3.4.2 Theory-Driven Model Fitting

SI Table S4 summarizes the best fitting model coefficients from the training dataset for each site and each CF. Across sites, the

$PM_{2.5}$ regression coefficient ($\alpha$) does not vary substantially: about 14% for CF1. Hygroscopic growth regression coefficients ($\beta$) vary greatly from site to site for CF1 even within the same region, $\beta_{CF1}$ for Delhi is double that for Hamirpur, perhaps due to a higher abundance of hygroscopic species (Chen et al., 2022; Gani et al., 2019).

The lack of consistency in fit is reasonable, as the Plantower proprietary algorithm and underlying physical-optical design of nephelometers means the sensor does not explicitly account for the underlying aerosol size distribution and composition.

The resulting datasets are therefore somewhat divorced from the expected pattern based on $\kappa$-Köhler theory. The ALT dataset removes the proprietary ATM correction, as well as assumptions of particle density present in the CF1 data, resulting in more consistent $\beta$ intra-regional values, though with less consistent $\alpha$ values.

### 3.4.3 Model Evaluation

For the Delhi and Hamirpur sites, both located in the IGP region, 2 parameter ATM and CF1 models yielded the consistent

improvements from 1 parameter models, as summarized in SI Fig. S16 for Delhi and Hamirpur respectively. The CF1 models were consistently more accurate than their ATM counterparts in Hamirpur, albeit by about 1% NRMSE and less than 1% $R^2$. Conversely, in Delhi, the ATM models systematically outperformed CF1 models by about 1% NRMSE and $R^2$. As evident in Fig. 1, Hamirpur experiences overall lower mass loadings than Delhi. Consequently, the absolute difference between the two signals due to the Plantower piecewise function (SI Fig. 1) above about 35 $\mu g/m^3$ is likely less important in Hamirpur than in

Delhi, where mass loadings are consistently elevated.

The theory-driven hygroscopic growth correction consistently improved performance from the uncalibrated baseline data across sites and CFs by 12% for ATM and 60% for CF1, on average. In North India, the theory-driven model performs within about2% of the 1 parameter models, even outperforming the 1 parameter ATM model in Hamirpur by 4.3%.

However, since the Plantower PMS5003 is a nephelometer, the signal should not necessarily follow the expected non-linear

hygroscopic growth with increasing RH above 60% as expected from a size resolved measurement technique (Crilley et al., 2020; Hagan and Kroll, 2020). As a result, the 2 parameter ATM and CF1 models in the IGP, with their additive RH terms, outperformed theory-driven by at least 2-3%. In Bangalore, the ATM theory-driven model performance was comparable to





the ATM 2 parameter data-driven model (see SI Fig. S16). This contrast to the stark difference in performance between the two methods in North India is likely a result of the more stable meteorology and source mixture in Bangalore, leading to less

dynamic aerosol hygroscopicity.

Since CF1 data produces models as accurate as or more accurate than ATM models, has been validated in studies around the world, and does not feature the same non-linear behavior as the ATM channel, we recommend using CF1 for calibration in the IGP. In Bangalore, the ALT data maybe useful and warrants further study in similar environments, including across South India. From our results the CF1 data is suitable for deployment in Bangalore and provides uniformity in calibration guidance.

Additionally, the 2-parameter model (with RH as additive terms to $PM_{2.5}$) follows previous studies (Barkjohn et al., 2021; McFarlane et al., 2021; Zheng et al., 2018) across continents and aerosol regimes.

Given these findings, we selected the following multi-season correction equations (Eq. (2-4)) for Delhi, Hamirpur, and Bangalore respectively. Although relatively simple, our calibration models greatly improve the reliability of low-cost sensor data across aerosol regimes. SI Figure S16 summarizes model accuracy, with NRMSE improvements from uncalibrated data

ranging between 5-20%. Figure 2 summarizes each model's bias in at each collocation site with seasonally, and diurnally segregated residuals. Across all sites, monthly bias of the calibrated data is within ±25%, in contrast to the uncalibrated data. SI Figure S17 additionally explores the residual structure and demonstrates the value of the selected model forms at reducing bias due to RH, and mass loading factors. The selected calibration equations reduce median bias to near 0% across sites, from median bias as high as 150% using the uncalibrated data at RH > 60%.

$$C = 0.546 \times CF1 - 93.6 \times RH + 50.3 \text{ (Delhi)} \tag{2}$$

$$C = 0.496 \times CF1 - 29.6 \times RH + 22.0 \text{ (Hamirpur)} \tag{3}$$

$$C = 0.515 \times CF1 - 13.9 \times RH + 14.1 \text{ (Bangalore)} \tag{4}$$

### 3.5  Model Evaluation

#### 3.5.1  Temporal Sensitivity

To identify the stability of the model and its parameters, we computed the 4-week ROLS for each of our selected models and compared performance to all other 4-week moving ROLS models. Each model's NMBE across time is shown in Fig. 4, where the grey squares in the top panel indicate less than 50% data completeness. Additionally, the bottom panel of Fig. 4 tracks the distribution of the diagonal of the matrices present in the top panel of the figure. Across sites, the choice of calibration period greatly changes the performance of the regression throughout the rest of the dataset and influences the selection of regression

coefficients. SI Figure S18 additionally explores the absolute bias, demonstrating that Eq. (2-4) bias are centered near 0. SI





Figure S19 illustrates the same analysis with NRMSE, showing that monthly ROLS model performance is generally stronger than the annual model within the training month, but rapidly deteriorates.

In Delhi, model performance and coefficient selection exhibit a seasonal pattern, with post-monsoon, and winter month models (Jan, Feb, March, Sept, Oct, Nov, Dec) performing well and selecting similar regression coefficients even across years (SI Fig. S20). When evaluating model performance on data within the same season, NRMSE is typically below 30% and $R^2$ is above 0.7. However, the post-monsoon and winter models perform poorly when evaluated on pre-monsoon data (March, Apr), with NRMSE exceeding 100% and $R^2$ falling below 0.1. For even the best performing pre-monsoon models, NRMSE rises above 50% during the pre-monsoon period data and above 70% for other seasons. Monsoon models (May, June, July, August) also lack transferability to other seasons, but perform well when evaluated on data from the same season (NRMSE < 30%).

The Hamirpur ROLS results are like those of Delhi, but over a shorter period and with more robust summer performance. The pre-monsoon models fit the largest magnitude $PM_{2.5}$ regression coefficient and fail to perform well (NRMSE > 50%) both within their own seasons' data and across other seasons' data. All other windows perform well (NRMSE $\leq$ 25%, $R^2 \geq$ 0.9) within their training window and across all other non-pre-monsoon test windows. The regression coefficients stabilize ($\beta_{PM_{2.5}}$ $\approx$ 0.5, $\beta_{RH} \approx$ -25), resulting in less seasonally variable model performance than in Delhi. Most likely the Delhi's models less robust performance across seasons relative to the Hamirpur model's performance is due to the broader diversity of sources in Delhi, making it more difficult to constrain the uncertainty due to factors including hygroscopic growth and particle size distribution.

Bangalore and Hamirpur results are similar in that both models are relatively stable and transferable across seasons. Bangalore model performance degrades and features less season-to-season transferability in the monsoon season months (July and Aug) but features accurate performance (NRSME < 20%) for the other seasons. Regression coefficients in Bangalore are relatively consistent even, albeit with more spread during the pre-monsoon.

Although model results and calibration formulation differ across sites, the temporal sensitivity analysis reveals several key lessons. First, there is no "free lunch" or universal model. Rather, aerosol and meteorological regimes shift rapidly, leading to underfit for annual models or overfit for seasonal models. Since annual models use data from across the distribution of aerosol compositions and size distributions, they generally perform within 5% of monthly models (SI Fig. S21). Outliers can be especially concerning at the physical limitations of nephelometers such as during Pre-Monsoon dust storms or the extremely humid Monsoon. Therefore, models trained within one single monthlong period do not necessarily transfer well to the next month, even within the same season and model feature selection. Consequently, we recommend calibration procedures in India and other similar environments maintain a long-term collocation with at least one LCS-FEM pair after the initial collocation period in the region of interest.

### 3.5.2 Spatial Transferability

Due to proximity and similarities in climate and aerosol characteristics, and since data-driven models from both IGP sites share the same parameters (CF1 and RH), we hypothesized that Delhi and Hamirpur models may be transferable. Figure 5 summarizes the relevant performance metrics with respect to spatial calibration transferability. The Hamirpur dataset performance





weakened after applying the Delhi model ($R^2$ decreased to 0.82, NRMSE increased to 39%) but still outperformed uncalibrated CF1 data. The Delhi dataset performance also weakened after applying the Hamirpur model ($R^2$ decreased to 0.78, NRMSE increased to 35%), a relatively modest performance degradation. From this exercise, we understand that although $PM_{2.5}$ is highly variable in the IGP, there may be enough of a "fingerprint" in aerosol characteristics from the background site that a single calibration equation could provide adequate performance improvement. However, a local calibration can provide performance improvements due to fine scale $PM_{2.5}$ variability unique to urban environments, especially for a megacity like Delhi.

Applying the North India models to the Bangalore test dataset resulted in contrasting performance, with an NRMSE of 71% and 24% from the Delhi and Hamirpur models respectively. It is likely the largely regional aerosol from Hamirpur has enough overlap in speciation and mass concentration range with the Bangalore aerosol that the models are somewhat interchangeable. This hypothesis is additionally evidence from the overlap in coefficients from the theory-driven hygroscopic growth equations. Clearly, the uniqueness of the Delhi aerosol prevents exchange between Delhi and Bangalore models, but with enough preserved from the regional contribution to support some support from the Hamirpur model to the Delhi data (although not from a Delhi model to Hamirpur data).

Some calibration efforts have sought a unified continental model for LCS by combining multiple reference-LCS pairs into one regression model (Barkjohn et al., 2021). Other studies have focused on interpolating between calibration sites to avoid washing out local effects, typically in a dense sensor network (Zheng et al., 2019). Our results show that although there are overarching similarities in model parameter selection, urban and rural environments are heterogeneous to the point of potentially barring a unified model. Additionally, seasonal variability within India necessitates at least monthly updates to model coefficients.

## 4    Conclusions

We collocated LCS with reference grade $PM_{2.5}$ monitors in three environments in India, two urban (Delhi, and Bangalore) and one rural (Hamirpur) over the course of multiple seasons to characterize LCS performance across shifting emissions and meteorological regimes and develop calibration models. Internally, PA-II units demonstrated strong consistency, with low intra-sensor bias and high correlation. Relative to reference instruments, uncalibrated sensor performance varied diurnally and seasonally with shifts strongly associated with extreme mass concentrations, RH, and coarse mode particles. The LCS signal generally overestimated mass concentrations relative to the reference instruments, a trend observed in literature to be associated with hygroscopic growth (Jayaratne et al., 2018; Malings et al., 2019). We also identified several dust storm episodes where the LCS signal underestimated by a factor of 2 – 6×.

We showed a relatively simple multilinear regression model using only the LCS $PM_{2.5}$ signal and LCS RH could produce results well correlated ($R^2 \geq 0.8$) with the reference signal at each site. These site-specific models provide the basis for a computationally efficient, well constrained (NRMSE $\leq$ 25%), and scalable calibration approach for low-cost sensing in India, despite the non-stationary and diverse aerosol dynamics of the region. Furthermore, we showed our models can be transferred



from site to site and still improve performance above the uncalibrated baseline, although a site-specific model generally has superior performance.

Our work also highlights a key caveat to LCS deployments and calibration in India, especially long-term deployment. Models trained at a site with only data from one season may perform more accurately within that season than a seasonally balanced model but are unreliable at other times of the year. Our results showed that this is especially important given the contrast in meteorology and mass concentrations between the Pre-Monsoon and Monsoon seasons. Although a multilinear regression approach produces well constrained results, these models are not transferrable among seasons. Therefore, we advise

future deployments to continuously operate a collocation site with at least one LCS-Reference pair to evaluate calibration drift. Accounting for the temporal and spatial dynamics of aerosol characteristics allow of the rapid scaling of LCS for communities in India to communities in need of transparent and accurate data.

*Data availability.*   Hourly concentrations for BAM-1020/1022 $PM_{2.5}$ as well as all PurpleAir $PM_{2.5}$ channels (CF1, ATM and ALT), and PurpleAir meteorological data (Relative Humidity, Temperature, and Dew Point) used in this study are available via the <insert data repository>.

*Author contributions.*   JSA, RKP, SV, MK, SG, SS, JG, and MJC designed the study. SV, HRM, MK, PA, AU, NB, SS, JG, and MJC carried out the data collection. MJC carried out the data processing and analyses. All coauthors contributed to interpretation of results, writing, and reviewing the paper.

*Competing interests.*   The authors declare that they have no conflict of interest.

*Acknowledgements.*   We are grateful to Open Philanthropy and The University of Texas President's Award for Global Learning for their
support. We are thankful to the US Embassy in New Delhi, the Center for Study of Science, Technology and Policy in Bangalore, and IGP-CARE in Hamirpur for institutional support.



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

**Figure 1.** Diurnal profiles of mean hourly seasonal BAM (reference) and uncorrected PA PM2.5 signals for Delhi, Hamirpur, and Bangalore using the CF1 channel. The number of valid hourly averages (quality assured according to methods outlined in Sect. 2.4 and summarized in Sects. 3.1 - 3.3) in each dataset is presented in the bottom left of each subplot. Winter (January, February), Pre-Monsoon (March, April, May), Monsoon (June, July, August, September), and Post-Monsoon (October, November, December).

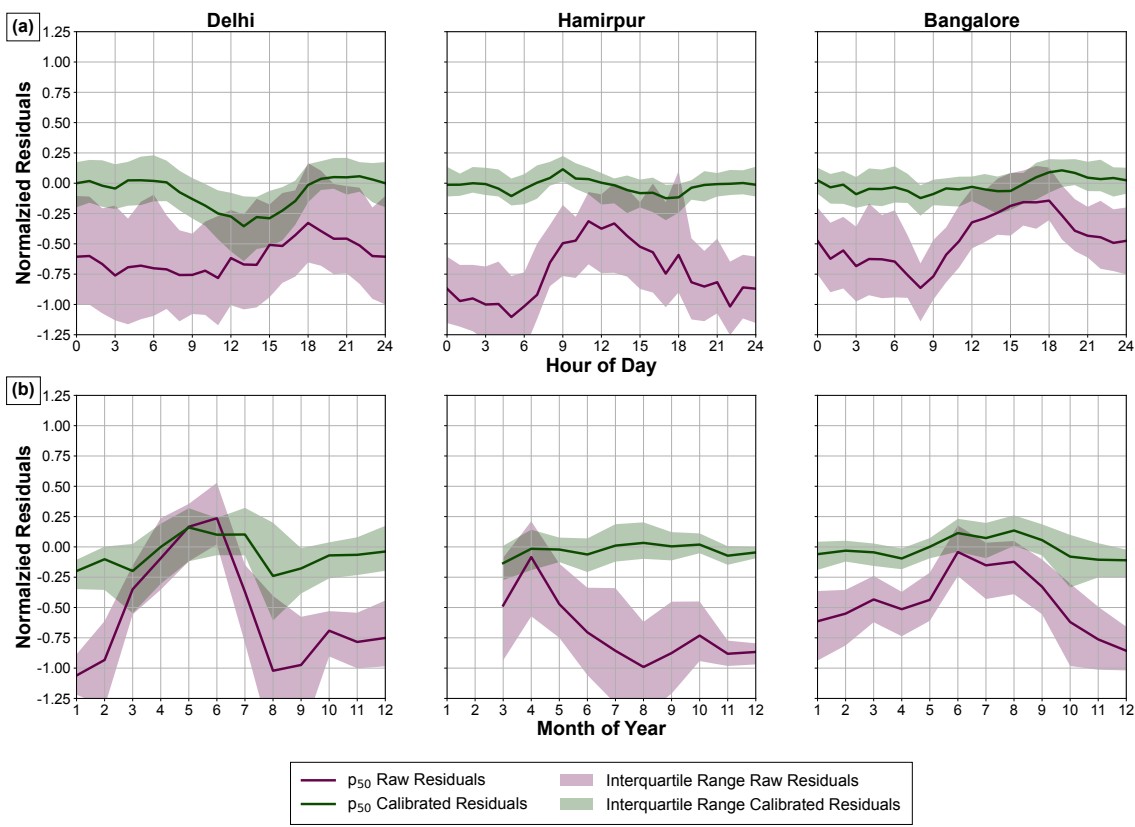

**Figure 2.** Normalized Residual distributions for the uncalibrated PurpleAir data (CF1) and the Calibration Models for each site. Bold lines represent the median ($p_{50}$) of the distribution, while the shaded area represents the interquartile range ($p_{75} - p_{25}$). The panel (a) shows the diurnal distribution, while panel (b) shows the normalized residual distribution binned by month. Compared to the residual distribution for uncalibrated (raw) data, the calibration effectively eliminates most seasonal and diurnal bias.

**Figure 3.** Scatter plots of the best performing 2 parameter hourly models for each of the sites in panel (a), with the corresponding normalized model residuals in panel (b) segregated by season and segregated by time of day in panel (c). In panel (a), sold line represents unity. In panels (b) and (c), dashed line represents normalized residual value of 0. In comparison to Fig. 2a, the normalized diurnal residuals in panel (c) are presented over a restricted y-axis, accentuating the residual structure.

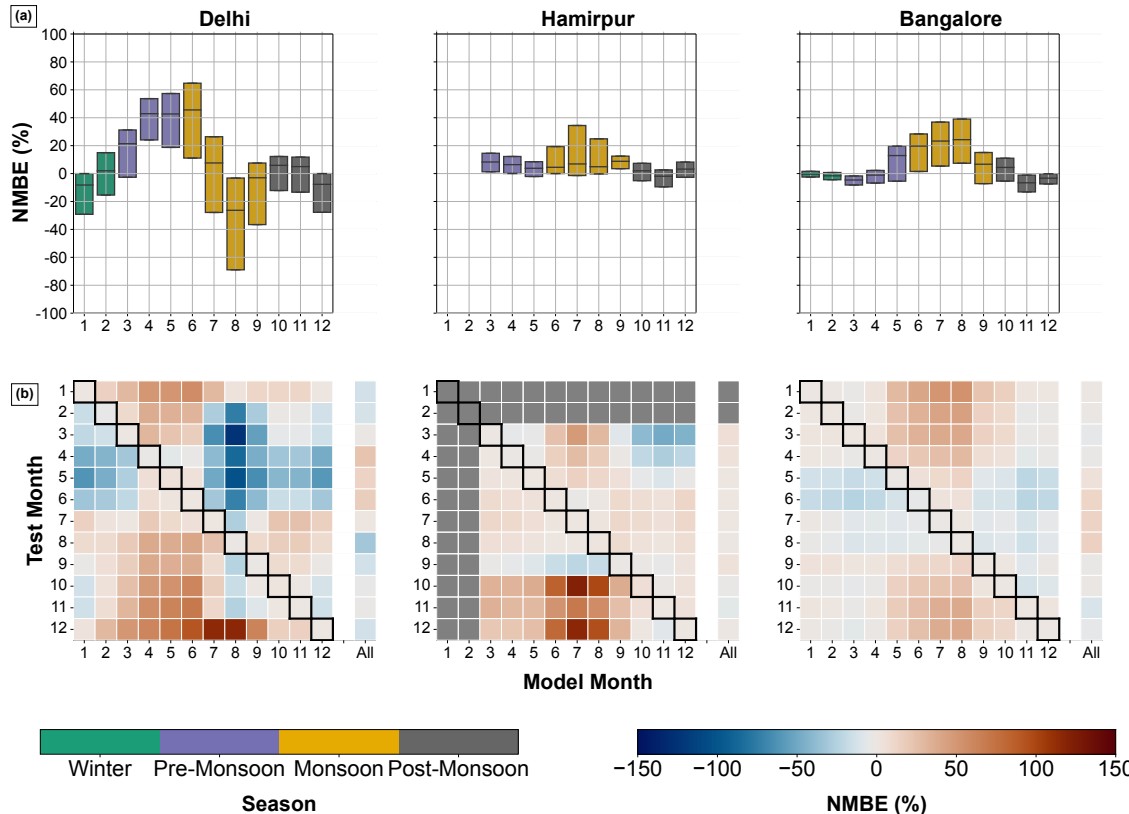

**Figure 4.** Panel (a) depicts box plots of the distribution of Normalized Mean Bias for a given model starting month of a 4-week rolling ordinary least-squares (ROLS) model on all other windows. The bottom, solid line and top of the boxes represent the $25^{th}$, $50^{th}$, and $75^{th}$ percentiles respectively. Panel (b) presents the median Normalized Mean Bias Error (NMBE) of a 4-week ROLS model trained starting in the month (colored by season) on the x-axis and evaluated on all other windows as binned by starting month on the y-axis. Gray boxes represent months without sufficient data. Models trained in the Pre-Monsoon (April, May) underpredicted in other seasons, contrary to the typical pattern of overprediction – this pattern is consistent at Delhi and Hamirpur on a normalized basis. Due to the more consistent meteorology and aerosol regime in Bangalore, there is more consistent performance regardless of training window. Finally, as a point of comparison, we present the performance of our long-term calibration in individual months at each site in the column in (b) titled "All." Consistent with our observation that 4-week models trained in a single month generally do not perform as well in other months, we also note that in general, monthly models perform somewhat better for a given month than the long-term model does. SI Figs S19 and S20 explore the MBE and NRMSE of this data.





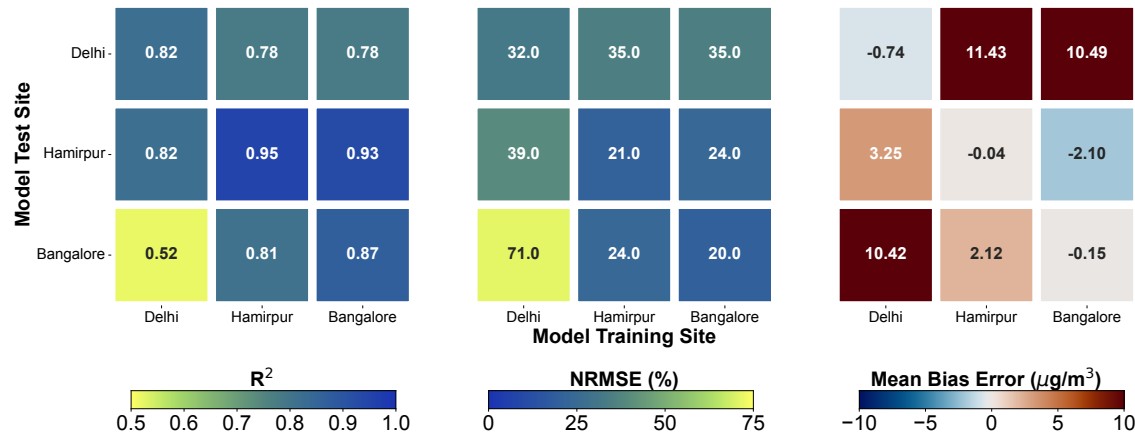

**Figure 5.** Performance evaluation metrics of Eq. (2-4) with the training site on the x-axis and the test-site on the y-axis. Metrics are Coefficient of Determination ($R^2$) (left), Normalized Root Mean Square Error (NRMSE) (center), and Mean Bias Error (right). For each metric, the diagonal pattern of best performance from upper left to lower right illustrates how calibration models perform best in the locations where they are trained. This finding illustrates how regional differences in meteorology and aerosol composition can limit the transferability of calibration relationships. Of note, the calibration model trained in Delhi performed quite poorly in Bangalore.



| | Delhi | Hamirpur | Bangalore |
|---|---|---|---|
| **BAM 1020/1022 PM$_{2.5}$ ($\mu$g/m$^3$)** | | | |
| p$_{10}$ | 23.0 | 10.6 | 10.8 |
| p$_{25}$ | 39.0 | 17.9 | 15.3 |
| p$_{50}$ | 71.0 | 34.3 | 21.8 |
| p$_{75}$ | 142.0 | 67.4 | 30.8 |
| p$_{90}$ | 237.0 | 125.4 | 42.1 |
| **PA-II CF1 PM$_{2.5}$ ($\mu$g/m$^3$)** | | | |
| p$_{10}$ | 31.0 | 18.1 | 12.7 |
| p$_{25}$ | 52.2 | 31.3 | 18.1 |
| p$_{50}$ | 117.0 | 63.4 | 31.0 |
| p$_{75}$ | 243.0 | 124.3 | 52.9 |
| p$_{90}$ | 375.0 | 218.0 | 74.5 |
| **PA-II ATM PM$_{2.5}$ ($\mu$g/m$^3$)** | | | |
| p$_{10}$ | 30.4 | 18.1 | 12.7 |
| p$_{25}$ | 43.0 | 30.2 | 18.1 |
| p$_{50}$ | 83.8 | 47.0 | 30.0 |
| p$_{75}$ | 180.0 | 82.7 | 42.4 |
| p$_{90}$ | 285.0 | 146.7 | 51.3 |
| **PA-II RH (%)** | | | |
| p$_{10}$ | 20.6 | 29.8 | 34.2 |
| p$_{25}$ | 29.6 | 44.1 | 48.0 |
| p$_{50}$ | 41.0 | 62.9 | 62.9 |
| p$_{75}$ | 48.5 | 77.4 | 73.8 |
| p$_{90}$ | 53.1 | 85.7 | 78.6 |
| **PA-II Temperature (°C)** | | | |
| p$_{10}$ | 18.7 | 17.2 | 23.7 |
| p$_{25}$ | 22.8 | 24.5 | 25.0 |
| p$_{50}$ | 29.2 | 30.6 | 27.7 |
| p$_{75}$ | 35.1 | 35.3 | 32.3 |
| p$_{90}$ | 39.0 | 40.3 | 37.1 |

**Table 1.** Summary of campaign measurements (quality assured according to methods outlined in 2.4 and summarized in 3.1 - 3.3) including 10[th] percentile (p$_{10}$), 25[th] percentile (p$_{25}$), 50[th] percentile (p$_{50}$), 75[th] percentile (p$_{75}$), and 90[th] percentile (p$_{90}$) for the campaign periods (Delhi: 2018 July – 2020 April, Hamirpur: 2020 January – 2021 January, Bangalore: 2019 June – 2020 August)





|  | CF1 | ATM | ALT |
|---|---|---|---|
| | $PM_{2.5}$ | $PM_{2.5}$ | $PM_{2.5}$ |
| *Delhi* | RH | $RH^2$ | $RH^2 \times T$ |
| | $PM_{2.5} \times RH$ | $PM_{2.5} \times RH^2$ | $PM_{2.5}^2 \times D$ |
| | $PM_{2.5}$ | $PM_{2.5}$ | $PM_{2.5}$ |
| *Hamirpur* | RH | RH | $PM_{2.5} \times RH \times T$ |
| | $RH^3$ | $RH^2$ | $PM_{2.5}^2 \times D$ |
| | $PM_{2.5} \times T$ | $PM_{2.5}^2 \times T$ | $PM_{2.5}$ |
| *Bangalore* | $PM_{2.5} \times T \times D$ | $PM_{2.5}^3$ | $PM_{2.5} \times RH^2$ |
| | $PM_{2.5}^2 \times T$ | $PM_{2.5}^2$ | $PM_{2.5}^2 \times T$ |

**Table 2.** Most relevant parameters selected through Sequential Feature Selection for each PurpleAir $PM_{2.5}$ channel by site: CF1 ("uncorrected" PurpleAir $PM_{2.5}$), ATM ("atmospheric corrected" PurpleAir $PM_{2.5}$), and ALT ("alternative" PurpleAir $PM_{2.5}$ – reconstructed from modeled size distribution data). Parameters: Relative Humidity (RH), Temperature (T), and Dew Point (D).