# Peer review of "Seasonally Optimized Calibrations Improve Low-cost Sensor Performance: Long-term Field Evaluation of PurpleAir Sensors in Urban and Rural India"

_Atmospheric Measurement Techniques, 2023_

## Author Comment (AC1)

**Reviewer 1:**

**Comments:**

We would like to thank Dr. Subramanian for his comments and transparency.

Disclosure: I will be joining CSTEP in July, where two coauthors were working during this study (and one continues to be part of CSTEP and will be reporting to me). CSTEP is also one of the three sites in this study.

Overall, this is a thorough study on LCS performance and correction at three different sites across India, nicely presented. My comments/questions are mostly minor. However, I am not sure whether the title is entirely correct - monthly corrections aren't a great improvement and don't even always carry over to other months of the same season. The final recommendation is to collocate at least one sensor with a reference for the entire study duration. As for site-specific, the Hamirpur and Delhi corrections appear reasonably interchangeable (see comment 20).

With feedback from all reviewers, we have decided to remove the phrase "site-specific" from the title, in line with the more recent direction in literature focusing on calibration time-periods (Levy Zamora et al 2023)

> Seasonally Optimized Calibrations Improve Low Cost Sensor Performance: Long-term Field Evaluation of PurpleAir Sensors in Urban and Rural India

Comments:

1. Abstract uses both Pearson r and $R^2$; please use one or the other for consistency. The $R^2$ values for "raw" (I call it uncorrected) data (0.55-0.74, Fig S16 - which show the final corrections in much better light!) show sensor performance that is not as impressive as r >= 0.9. Incidentally, the Pearson r result only appears in the abstract and not in the main text.

   For the purposes of consistency, we have amended the text to uniformly use the Coefficient of Determination ($R^2$) rather than Pearson's r.

   > Without calibration, the PA-IIs were moderately well correlated with the reference signal (R2: 0.55 - 0.74)…

2. Showing a table of fit statistics for the uncorrected and final corrected data (and maybe the spatial transferability results) in the main text would improve clarity. Currently, these key results are discussed in the text but only presented graphically in Figure S16.

   Thanks for this suggestion. We have improved clarity by promoting Figure S16 to the main text as the new Fig. 3.  Upon transferring the figure, we identified a transcription error in the NRMSE matrix resulting in the incorrect numbers displayed in the Bangalore panel as well as minor rounding issues in the other panels. This correction has no impact on interpretation.

3. Line 43: The Plantower sensors do sense particles above 0.8 micron, even up to 2 micron - just not very efficiently. Kuula et al. say 0.8 micron but based on "Valid detection ranges...defined as the upper half of the detection efficiency curve" - which seems different from "failing to characterize". Maybe something like "do not adequately characterize".

We agree and have rephrased.

"…do not adequately characterize fine particles above 0.8 microns…"

4. From Wallace et al. (2021): "The ALT method is based on the number of particles per deciliter reported by the PMS 5003 sensors in the PurpleAir instrument for the three size categories less than 2.5 μm in diameter." It is unclear that these are independent measures; Kuula, He/Dhaniyala, Ouimette, Andy May, and others have shown the size distribution is not real. Since the ALT method isn't finally used, maybe move these results to SI and improve clarity by focusing on the two metrics (CF1 and ATM) that most people use anyway.

We agree that the citations in the reviewer's comment are strong evidence against treating the size resolved data from Plantower sensors as true size distribution measurements. Nonetheless, since publications with the ALT method have grown in popularity and is featured on the PurpleAir map, we sought to highlight negative findings as evidence against its application. Therefore, we believe it would be best to keep these findings in the text as a point of contrast to the CF1 and ATM data. We have increased the level of detail in which we describe the differences between CF1, ATM, and ALT as well as provided a brief justification for reporting calibration results for each data channel.

However, the particle number data is known not to reflect the actual ambient size distribution since the Plantower PMS5003 is not a particle sizing instrument, but rather reflects a modeled size distribution using assumptions for relationships between size bins that is not always accurate for atmospheric conditions (Ouimette et al., 2021; Hagan and Kroll, 2020; He et al., 2020; Kuula et al., 2020). SI Figure S1 shows the ALT to CF1 ratio is approximately 0.15:1. Although the CF1 and ATM data have dominated most calibration efforts (Malyan et al., 2023; Puttaswamy et al., 2022; Barkjohn et al., 2021; McFarlane et al., 2021; Magi et al., 2020; Malings et al., 2019), the usage of ALT data continues to propagate in peer-reviewed literature (Wallace and Zhao, 2023; Wallace and Ott, 2023). Therefore we use CF1, ATM, and ALT in our study to work towards harmonizing a calibration approach for PA-II in India.

5. Line 83: "while can the BAMs provide" should be "while the BAMs can provide"

We have fixed this grammatical error.

6. Line 150: Instead of "block averaged", recommend using "hourly averages of" - because I think that's what is being done. "Block averaging" is not otherwise clarified in the manuscript and "hourly averaging" is easily understood.

We have reviewed the entirety of the text and replace "block average" with "hourly average."

7. Line 160: "the quotient of the mean and standard deviation" seems the inverse of the CV - might relative standard deviation be easier to understand?

Thanks for catching this error in our text. We reviewed our code pipeline and ensured that we employed the Coefficient of Variation as it is conventionally defined: $CV = \frac{\sigma}{\mu}$, therefore this is a typo in the text. We have rephrased.

   "… the quotient of standard deviation and mean…"

8. Eq. 1 is an unusual formulation, so perhaps the original study that used this formulation (as far back as I can track it!) should be cited?
   1. Zhang et al. (1994) https://doi.org/10.1080/1073161X.1994.10467244 (Their Figure 4 was used in a workshop report Laulainen et al. 1993 that was then cited by Chakrabarti et al. 2004.)

We recognize by skipping the exact derivation of our form, we may have omitted some key details on how Eq. 1 is related to the form cited in Chakrabarti et al 2004. We have added this section to the SI:

The correction equations used in Laulainen et al. 1993 and Chakrabarti et al. 2004 take the form:

(1) $CF = 1 + 0.25\frac{RH^2}{1-RH}$

(2) $C_{corrected} = \frac{C_{raw}}{CF}$

Where RH represents the fractional RH, expressed on scale of $0 - 1$, CF represents the hygroscopic correction factor, $C_{raw}$ represents the light scattering instrument PM mass concentration, and $C_{corrected}$ represents the PM mass concentration corrected for hygroscopic growth. Combining the two equations yields:

(3) $C_{corrected} = \frac{C_{raw}}{1+0.25\frac{RH^2}{1-RH}}$

From Laulainen et al. 1993, the selection of 0.25 in the denominator of (3) was found to vary with chemical composition, and in fact suggests a value of 0.328 at a site dominated by ammonium sulfate. As speciation across India is known to be strongly variable dependent based on seasonal and diurnal factors, we instead allowed for a best-fit approach described in the text to select the best fitting factor yielding the following:

$$(4)\ C_{corrected} = \frac{C_{raw}}{1+\beta\frac{RH^2}{1-RH}}$$

Furthermore, given that (1) was derived for use with integrating nephelometers, we chose to include an additional term ($\alpha$) in the numerator to account for the differences in instrumentation of the truncated nephelometer. Therefore, we arrived at:

$$(5)\ C_{corrected} = \frac{\alpha \times C_{raw}}{1+\beta\frac{RH^2}{1-RH}}$$

Finally, we replaced $C_{raw}$ with P to represent any PM$_{2.5}$ mass concentration signal (CF1, ATM or ALT) from the PurpleAir, and simplified C$_{corrected}$ to C.

$$(6)\ C = \frac{\alpha \times P}{1+\beta\frac{RH^2}{1-RH}}$$

9. Line 216 says the rolling OLS performance was compared against other two-week periods, but the results suggest monthly evaluations. Please clarify.

This is a typo from an earlier iteration of the analysis and has been corrected from "2-week periods" to "4-week periods."

10. Lines 263-264: 15% and 14% seem not that different to warrant an explanation.

We agree, this sentence has been abridged for conciseness.

The CV test removed about 15% from each site.

11. Lines 267-268: Why is Delhi so unusually lossy? Both IGP sites are significantly lossier (data recovery <40%) than the CSTEP site (75%), which is surprising and not well explained given e.g. my comment #11 that even a 1% difference in the results was explained even if not seemingly necessary.

We agree it is necessary to offer summary comment on the dataset losses. First, the BAM in Bangalore failed less within the collocation periods than the BAMs in the IGP (Delhi and Hamirpur. Also, the CSTEP site in Bangalore was much better staffed allowing for fewer gaps due to power or Wi-Fi fluctuations, and maintenance such as BAM tape replacement. We have added these details to the end of the paragraph.

The smaller number of data points available for the Delhi and Hamirpur sites principally arose because of relatively more downtime of the BAM instruments at these two locations.

12. Lines 280-282: This is unclear from the figure, which I interpreted as "the PA line is mostly above or close to the BAM line for the pre-monsoon period at Hamirpur; it underestimates about half the time for Delhi, maybe."

We agree this phrasing requires some streamlining and have accepted similar language to the reviewer's suggested wording.

13. Line 282: coarse aerosol are particles larger than 2.5 micron. $PM_{2.5}$ is called fine PM, e.g. https://www.epa.gov/pm-pollution/particulate-matter-pm-basics

We agree with the spirit of this comment and have rephrased the sentence for clarity, as presented below.

Here, we intended to refer to the fact that the coarse-*mode* of a particle size distribution typically is not monodisperse, but rather includes a lower tail of ~ 1-2.5 μm sized particles that contributes to $PM_{2.5}$ mass. While this is not generally a major contributor to $PM_{2.5}$ loadings under ordinary conditions, dust storms or other events that produce predominantly coarse aerosol can lead to elevated $PM_{2.5}$. See, for example, the archetypal size distributions of Seinfeld and Pandis 2016 (3rd Edition of Atmospheric Chemistry and Physics) Figure 8.10, which illustrate that while the median of the coarse mode is >2.5 microns, the tail of the coarse mode extends well below 2.5 microns.

Although both the Delhi and Hamirpur sites feature relatively low bias in the pre-monsoon period, they underestimate mass concentrations in this season, perhaps due to the influence of wind-blown mineral dust, as observed elsewhere in field and lab evaluations…

While crustal material does not generally dominate $PM_{2.5}$ mass, during dust storms the lower tail of the coarse mode aerosol can lead to substantially elevated $PM_{2.5}$ concentrations in India.

14. Lines 297-301: I appreciate this decision. Good call.

Thanks, glad you agree!

15. Line 306: I don't think T & dewpoint coefficients were reported anywhere. Are these statistically different from zero?

While the coefficients are not statistically different from zero, adding terms with temperature and dew point only resulted in marginal improvements ($\Delta R^2 \approx 0.01$). Therefore, we have clarified this comment by changing the phrasing.

Temperature and dewpoint terms only imparted marginal improvements to calibration models ($\Delta R^2 \approx 0.01$), and it is not determinable if the models are deriving a spurious correlation or detecting underlying aerosol or instrument properties.

16. Lines 326-327: Text is unclear ("CFs"??) and not presenting these key results in the main text (ideally as the table requested earlier) is not helping.

We agree the text requires clarification. In addition to adding SI Fig S16 to the main text we have rephrased the sentence.

The theory-driven hygroscopic growth correction consistently improved performance from the uncalibrated baseline data across sites by 12% for ATM and 60% for CF1, on average (Fig. 3).

17. Fig S16 uses "RHC" to indicate the theory-driven fRH-esque approach, which is odd. What does RHC stand for? Maybe just say "theory".

We have reviewed the entirety of the text and replaced "RHC" with "theory-driven."

18. The discussion of Fig S16 doesn't align with the results shown. The Bengaluru theory-driven performance has the same $R^2$ as the 1-parameter model, lower than the 2-parameter model. NRMSE is lower for theory model than any of the empirical models. (One might quibble about differences of 0.02 and argue that is "comparable", but the surrounding text touts differences of 2-3% so...)

We agree this text should be more consistent. In addition to promoting SI Fig S16 to the main text (as the new Fig. 3), we have harmonized the analysis of the results it illustrates to clarify we identified differences of more than 3% NRMSE or $R^2$ as clearly more robust. Therefore, we have rewritten this section to state the theory-driven models are more less robust than the data-driven in North India (Delhi, Hamirpur), and offer only marginal benefits in Bangalore. Therefore, we choose to uniformly apply a data-driven model.

"…the 2 parameter CF1 models in Delhi and Hamirpur, with their additive RH terms, outperformed theory-driven by at least 3%. In Bangalore, the theory-driven model performance was comparable to the data-driven models (about 1% NRMSE, see Fig. 3). This contrast in performance between the two methods in Delhi and Hamirpur is likely a result of the less seasonally variable meteorology and source mixtures in Bangalore, leading to less dynamic aerosol hygroscopicity."

19. Eq 2-4 - is RH used as a fraction in these equations? Please specify. RH is usually reported as % and can be used directly in such equations, so maybe use that convention instead.

To maintain consistency across data-driven and theory-driven approaches we choose to use the fractional form. For consistency with other models from literature, we have re-written these calibration relationships in terms of %.

20. Line 408: Unclear if this parenthetical is really the case. Applying Hamirpur correction to Delhi or vice-versa produces relatively similar $R^2$ and NRMSE 0.82/39% or 0.78/35% with no clear winner.

We agree the is not strong enough to fully support the assertion and have removed the phrase.

Clearly, the differences in composition of the Delhi and Bangalore aerosols prevents exchange between models at these two sites, but with enough preserved from the regional contribution to support some support from the Hamirpur model to the Delhi data.

21. Lines 422-423: Dust storms were not identified nor discussed elsewhere in the text. Dust is only hypothesized as a potential explanation for a result (lines 280-284).

We agree the language suggests stronger than presented findings on the influence of dust storms. We have rephrased.

"We identified periods of low-cost sensor signal underestimation by a factor of 2 – 6× in the Pre-Monsoon in Delhi and Hamirpur when supra-micron wind-blown dust particles are relatively abundant."

22. Lines 436-437 - check the sentence.

We will fix the grammar.

23. Data availability: insert data repository link.

We have setup a repository on Dryad and will add the link before final publication.

24. Vos et al. is almost three pages of authors for one reference that doesn't actually contribute much to this specific manuscript. Can you just say "GBD 2019 Diseases and Injuries Collaborators" as the group is known on the paper?

We agree the Vos et al citation full form is unnecessarily large and have replaced it as recommended as well as Pandey et al.

25. Fig 4 caption is really long, but has a simpler explanation of the results than what is in the text...

We have refined the Fig. 5 (figure formerly referred to as Fig. 4) caption to describe the take-home message of the figure more concisely.

Assessment of inter-seasonal transferability of seasonal models. Panel (a) depicts box plots of the distribution of Normalized Mean Bias Error (NMBE) for a given model starting month of a 4-week rolling ordinary least-squares (ROLS) model on all other windows. The bottom, solid line, and top of the boxes represent the 25th, 50th, and 75th

percentiles respectively. Panel (b) presents the median NMBE of a 4-week ROLS model trained to start in the month (colored by season) on the x-axis and evaluated on all other windows as binned by starting month on the y-axis. Gray boxes represent months without sufficient data. Models trained in the Pre-Monsoon underpredicted in other seasons, contrary to the typical pattern of overprediction – this pattern is consistent at Delhi and Hamirpur. As a point of comparison, we present the performance of our long-term calibration in individual months at each site in column (b) titled "All." Consistent with our observation that 4-week models trained in a single month generally do not perform as well in other months.

---

## Author Comment (AC2)

**Reviewer 2:**

**Comments:**

We would like to thank the reviewer for their comments.

Overall, this paper is well-written and was completed in a systematic manner. The main drawback is it fails to identify and highlight innovation in the work. The stated aim of the paper was to "identifying robust calibration protocols", which feels like a step in completing the QAQC process. As a whole, the Plantower/purple-air pm sensor's accuracy and precision are well-studied. I recommend that the authors revisit the abstract and introduction in order to highlight the contribution to the scientific body of knowledge that this work provides.

We thank the reviewer for the opportunity to clarify our objectives and novel contributions. We have reworked the manuscript abstract accordingly.

Lower-cost air pollution sensors can fill critical air quality data gaps in India, which experiences very high fine particulate matter ($PM_{2.5}$) air pollution but has sparse regulatory air monitoring. Challenges for low-cost $PM_{2.5}$ sensors in India include high aerosol mass concentrations and pronounced regional and seasonal gradients in aerosol composition. Here, we report on a detailed long-time performance evaluation of a popular sensor, the Purple Air PA-II, at multiple sites in India. We established 3 distinct sites in India across land-use categories and population density extremes (North India: Delhi [urban], Hamirpur [rural]; South India: Bangalore [urban]), where we collocated the PA-II with reference beta-attenuation monitors. We evaluated the performance of uncalibrated sensor data, and then developed, optimized, and evaluated calibration models using a comprehensive feature selection process with a view to reproducibility in the Indian context. We assessed the seasonal and spatial transferability of sensor calibration schemes, which is especially important in India because of the paucity of reference instrumentation. Without calibration, the PA-II was moderately correlated with the reference signal ($R^2$: 0.55 - 0.74) but was inaccurate (NRMSE $\geq$ 40%). Relative to uncalibrated data, parsimonious annual calibration models improved PA performance at all sites (cross-validated NRMSE 20-30%, $R^2$: 0.82-0.95), and greatly reduced seasonal and diurnal biases. Because aerosol properties and meteorology vary regionally, the form of these long-term models differed among our sites, suggesting that local calibrations are desirable when possible. Using a moving-window calibration, we found that using seasonally-specific information improves performance relative to a static annual calibration model, while a short-term calibration model generally does not transfer reliably to other seasons. Overall, we find that the PA-II can provide reliable $PM_{2.5}$ data with better than $\pm$ 25% precision and accuracy when paired with a rigorous calibration scheme that accounts for seasonality and local aerosol composition.

Abstract:

I suggest strengthening the aims in your opening line since there are now a lot of calibration papers for the Plantower/purple air. I.e., what are you adding to this body of literature?

We agree and have strengthened the title as well as the abstract.

Seasonally Optimized Calibrations Improve Low-cost Sensor Performance: Long-term Field Evaluation of PurpleAir Sensors in Urban and Rural India

Clarify why these three are distinct and what they add to the study (e.g., urban, suburban, background, forested, etc.)

We agree these distinctions are important and have included the terms "urban" and "rural" in the abstract. We further describe each site in detail in the methods section and SI.

Can you clarify what a "major season" is?

We agree this term may be confusing and removed the line as it is extraneous.

It would be useful to the reader if you briefly state how the aerosol and meteorology vary by the site since I assume they capture unique environments.

We agree and have described them in the methods section.

The National Capital Region along with the rest of North India experiences dynamic meteorology with cold wet winters, warm drier post-monsoons and pre-monsoons, and hot wet monsoons (SI Fig. S4)… During the course of our campaign, Delhi experienced extreme $PM_{2.5}$ concentrations during the post-monsoon agricultural burning seasons and characteristic winter inversion layers, with a relatively low-pollution monsoon season consistent with expected seasonal trends…

…Although campaign median $PM_{2.5}$ concentrations at the site (Table 1) are high in the global context, this site's remote location outside of both cities and villages means that concentrations do not reach the same peaks as in Delhi. However, there are still many local sources of aerosol air pollution in rural North India such as biomass burning for cooking and heating…

In Bangalore, emissions are dominated by traffic and dust resuspension… Compared to Delhi and Hamirpur, winters are milder, and the climate is more consistent year-round in Bangalore (SI Fig. S6).

This sounds like a more innovative part of the work, expand?
We used a comprehensive feature selection process to create optimized site-specific calibrations.

We agree and have added relevant details in the methods section.

> To iterate across all possible arrangements of predictors - including additive terms, interaction terms, as well as polynomial terms up to order 3 – we implemented Sequential Feature Selection (SFS) using the Python package scikit-learn 0.24.2. SFS uses a greedy approach to converge on the best-performing model for a user-defined number of parameter (Raschka and Mirjalili, 2019; James et al., 2013; Ferri et al., 1994). For example, if a user wanted a 2-parameter model from a set of 10 features, SFS would iteratively compare 90 models, the set of all possible 2-parameter feature permutations, using a robust regression metric (such as adjusted $R^2$ or Bayesian Information Criterion [BIC]). In our approach, we first use SFS to define the best-performing n-parameter model starting with all possible parameters (n=34). We then compare adjusted $R^2$ across best-performing n-parameter models to measure the impact of model complexity. If increasing parameters results in only marginal improvements ($\Delta R^2 \approx 0.01$), then it is unnecessary to use those additional features. The overall most robust model, therefore, reflects both the best possible selection of features as well as feature parsimony.

Since the form varies by site, do you make a recommendation to other users on what to include in their model?

We agree this is valuable information for the community. From our analysis, we show that the calibration equation is very similar for our two sites in North India region, so in our results and conclusions we endorse using this form across distinct settings in this region. Furthermore, we demonstrate that the general form for North India only marginally degrades the best fitting parameters in Bangalore, so it is acceptable for usage based on our work. We discuss this point at length in the results and conclusions.

Can you clarify how it's "successful" if it does not work overall?

By "successful" we mean robust performance metrics within the training season. We have clarified the language to be clear we are referring to within-training season performance in this sentence.

> In contrast, we demonstrate that a short-term calibration exercise for one season with robust metrics within the season may not transfer reliably to other seasons.

Introduction:

Ln 39: Is "mischaracterize" the correct word here? I am confused by the goal of the statement.

We have edited to use more precise language.

> Optical sensors inaccurately estimate mass from aerosol scattering properties, since $PM_{2.5}$ is a mixture of particle sizes and chemical compositions thus resulting in spatial-temporal variability in optical properties…

Line 41-45 – This fails to cite enough work to support this statement. I suggest including more work from the past 2-3 years. This will also help you identify the innovative part of this work. Several LCS PM networks have thorough publications on calibration methods based on long-term field data.

We appreciate the rapid growth of research efforts in developing low-cost sensors networks, and have added more recent citations. It is worth noting the references in the first edition of the manuscript were strategically selected to emphasize efforts within India, given the difference in both aerosol regimes and infrastructure between the US (where a plurality LCS literature is based) and India.

The end of the introduction feels more like concluding remarks.

We recognize that this may be a matter of differing styles or tastes, but we believe that a brief paragraph summarizing of the approach and results in the introduction section is reasonable and provides a structure that improves the accessibility and readability of the rest of the paper. We'd welcome input from the AMT editors if they disagree with this stylistic choice.

Methods:

150: Please define "block-averaged" in the text.

We have reviewed the entirety of the text and replaced "block average" with "hourly average."

Ln 151: How do you determine what constitutes "imprecise points"?

 CV using the 6 nodes?

We calculated the Coefficient of Variation on all available Plantower signals, so for 3 collocated PA-II that would be the CV of 6 data points.

> … if we had three PA-IIs at a site, we averaged the six values together – two from each unit – to estimate a single data point. We established 80% completeness criteria (or 24 2-minute data points) for each hourly average, and at least 2 valid Plantower hourly averages for the resulting site PA datapoint. Imprecise site points were removed using the coefficient of variation (CV), the quotient of the standard deviation, and the mean of the collocated Plantower sensors for a given 2-min raw sample. CV values greater than 0.2 were removed, broadly consistent with approaches used by other studies…

Ln 160: "the quotient of the mean and standard deviation of the sensors" – How are these values were used?

We agree this point is unclear. There is a typo in the text, we have rephrased.

> We established 80% completeness criteria (or 24 2-minute data points) for each hourly average, and at least 2 valid Plantower hourly averages for the resulting site PA data point. Imprecise site points were removed using the coefficient of variation (CV), the quotient of the standard deviation, and the mean of the collocated Plantower sensors for a given 2-min raw sample. CV values greater than 0.2 were removed, broadly consistent with approaches used by other studies.

Ln 185: I think this is partially correct. Testing has shown that it exponentially overestimates at high RH, but these conditions are less likely to be sustained in real world environments.

We have adjusted our language accordingly.

> Although the theory-driven model should produce the most transferable models since theory should apply in all environments, the underlying data processing of the Plantower - a truncated nephelometer … may result in a bias structure better explained by a linear RH correction than an non-linear correction for the dynamic range of RH under real-world conditions.

Can you also provide local regulatory values in addition to WHO?

We agree this will add valuable context.

> While the annual average is low in comparison to cities in North India as well as the Indian National Ambient Air Quality Standard of 40 $\mu g/m^3$ it exceeds the WHO annual guideline value of 5 $\mu g/m^3$ and hourly winter concentrations often exceed 50 $\mu g/m^3$. Consequently, Bangalore has been designated for air quality improvement under the Indian National Clean Air Programme…

Please provide a reason for "We removed all raw PM2.5 data points outside of the range 5 – 500 µg/m3"

> Strongly contradicts "with peak daily (hourly) in excess of 500 µg/m3" line 103

From the specification sheet for the instrument as well as previous performance characterizations cited in text, we identified 5- 500 $\mu g/m^3$ as the operational range of the instrument on an hourly basis.  We acknowledge our assertion that the daily maximums in exceedance of 500 $\mu g/m^3$ was used as motivational text to emphasize the urgency of $PM_{2.5}$ air pollution in India could cause confusion with our later implementation of this quality assurance procedure. However, from our collocation study, applying this filter only removed about 1% of hourly data points from the raw dataset. Therefore, we have clarified the text by removing line 103.

How many minutes were required for the data to be averaged up to 1 hr? Is that the 80%?

Yes, the 80% refers to the number of 2-min PA-II data points required for a valid hourly average. We have clarified the text this corresponds to 24 datapoints/hour.

> We established 80% completeness criteria (i.e., 24 2-minute data points) for each hourly average…

You should cite the sources that influenced you to choose these covariates (e.g., ln 176, 188, etc.)

We have ensured the relevant sources are properly cited throughout the text.

Ln 215: A similar method was employed in "Identifying optimal co-location calibration periods for low-cost sensors." Compare results?

We agree there are some similarities in our approaches, we will ensure this paper is referenced, and used to contextualize our method. We agree with the findings of Levy Zamora et al 2023, that there are optimal collocation periods – in our case the Post-Monsoon – which are more transferrable than other periods. We have added these details to our results.

> Previously Levy Zamora et al. (2023) identified diminishing returns in improvements to calibration regressions after about 4 weeks of collocation in Baltimore, USA, if that period encapsulated a representative range of $PM_{2.5}$ and RH conditions. Here we build on this work by seeking to identify which 4-week period is ideal at our sites in India since annual median $PM_{2.5}$ concentrations at Delhi and Hamirpur sites are about 10× higher than Baltimore and reflect a different mixture of chemical composition and aerosol properties.

Ln 230. Please clarify "US EPA's data reduction process"

We have added a citation to the EPA SOP for maintaining and managing the BAM-1020 at embassy and consulate sites, which describes the process used by the State Department AirNow network in compliance with EPA protocols.

Ln 249: missing word?

We removed it to fix this grammatical error.

A general comment on the results: it is very acronym heavy, and I think it sometimes takes longer to mentally decrypt than it would be if it was just written out.

We have reviewed the text and clarified as necessary. For example, instead of referring to both the site and the location (ie, IGP-CARE vs Hamirpur) we have simply referred to the location.

Ln 363 – extra comma

We have fixed this grammatical error.

Ln 369 – Can you add the reasons for this trend here?

How do you think the notable differences in data likely influenced some of the transferability?

We add details to better describe this trend.

> Monsoon meteorological conditions contrast with other seasons – it is humid, windy, cloudy, hot, and frequently rains (SI Figs. S4-S6). These conditions result in lower emissions (i.e., less biomass burning for heating relative to winter), as well as act to suppress emissions (i.e., wet deposition) resulting in lower average seasonal mass concentrations in the Monsoon (SI Figs. S3 and S7). Consequently, models trained in the monsoon poorly translate to other seasons.

In my experience, LCS struggle more at high concentrations. Can you discuss concentration ranges more?

We explore this trend in SI Figure S17 panel a, which is referenced in the main text. While we do see performance degradation as mass concentrations increase, our calibration residuals are not sensitive to mass loading. We have added an additional sentence in Ln 347 around the discussion of SI Fig S17 and the impact of mass loading.

> The calibrated residuals distributions demonstrate marked improvements across the full range of mass concentrations, unlike the raw residuals which show increasing uncertainty at high mass concentrations.

How do you think the difference in complexities affected the transferability? Since they are quite different, they may be overfitting and that reduced the transferability too. It would be interesting to see something like Figure 5 all with the same model structures.

We agree model complexity likely impacts transferability. Figure 5 (Figure 6 in the new revised manuscript) already uses the same model structure for each model design according to equations 2-4. We have clarified this point in the figure caption.

> Performance evaluation metrics of Eq. (2-4) with the training site on the x-axis and the test site on the y-axis…

Figure 1: To clarify, is 779 the number of data points total (could be 10 points for 1 am and 100 points for 9 am) or the number for each hour? If it's the first, did you check that there are a comparable number of points per hour?

Yes, it is the total number of hourly averages. We appreciate the concern that some hours may not be represented properly, however, we found close to uniform coverage, with hourly totals contributing no more than ~ 7% to the total number of data points on a given plot, compared to the ideal ~ 4% (1/24). Therefore, we do not expect a systemic bias based on data availability of one hour in a season versus another. We have amended the figure caption accordingly.

No single hour of day represents more than about 7% of the total dataset shown in the bottom left corner of each plot.

Figure 1 would also be nice with the range highlights like Figure 2.

Thanks for this nice idea. We agree that including the range helps to better illustrate the statistical properties of the distributions, however in Figure 1, there is overlap in the diurnal profiles at some key periods (such as in Delhi, panel b). We experimented with adding this shading, but ultimately concluded that adding the shading for the distribution ranges added so much visual clutter that it would have made the figure less interpretable.

Figure 3 – Check this figure for visual accessibility.

Thanks for the suggestion. We have updated all our figures to ensure visual accessibility using the colormaps recommended by Crameri et al 2018.

Figure 5 – Is there a training/testing split in time for each site? I am confused as to why the site applied to itself changes.

The test-train split here is the same split used to derive and evaluate Eq. 2-4, such that the error metrics reflect evaluating each model from the y-axis exclusively on the test data described in the methods section, rather than on the training data used to generate each model. We have amended the figure caption accordingly (now called Figure 6 in main text).

At each site, we compute performance metrics by comparing the calibration model output to an independent test set that was held out from model training.

Conclusions:

Ln 416 - extra comma

We have fixed this grammatical error.

Ln 423 – Dust storms were not discussed with these numbers in the text. Please add a discussion so it is appropriate as a conclusion.

We agree the language suggests stronger than presented findings on the influence of dust storms. We therefore have rephrased.

> We identified periods of low-cost sensor signal underestimation by a factor of 2 – 6× in the Pre-Monsoon period at the Delhi and Hamirpur sites, when supra-micron wind-blown dust particles are relatively abundant.

Ln 430 – Some of the thoughts on seasonal and location transferability have been described elsewhere, but the discussion on the difference in the PM composition of the sites is interesting. Do you have thoughts on how the community can account for these differences if co-location is not feasible?

We agree that PM composition plays a key role in spatial transferability. We have added these relevant details to the text.

> Based on our analysis, we hypothesize that it is better to use a model developed at a background site such as Hamirpur to correct data from an urban environment such as Delhi, since the composition of PM in Hamirpur represents a good subset of the variability in Delhi. On the other hand, since there are PM species only found in some urban environments in India using models from these industrial microenvironments are less likely to produce reliable results outside of the training location.

---

## Author Comment (AC3)

**Reviewer 3:**

**Comments:**

We would like to thank the reviewer for their comments.

In this manuscript, the authors analyze standard BAM and Purple Air data from three sites in India to identify the optimum calibration procedures. The analysis may include some important and useful results on use of low cost sensor data, but at present, I had a hard time following the authors procedures and conclusions. There is some extraneous results in the manuscript that is distracting (like the theory driven calibrations that are not used) and too little clarity on what the authors did and how their calibrations performed compared to some of the standard published calibrations equations.

We thank the reviewer for asking us to streamline the main research questions we are trying to answer. We substantially edited the manuscript to address the general suggestions to refine our discussion, clarify our methods, and highlight our key contributions.

We agree that theory-driven results may seem distracting since they are negative findings, however, they remain an area of active investigation in the community, so we intend to keep them in the paper. Nonetheless we take this review's constructive feedback and valuable perspective seriously. We have added a section to the SI describing the derivation of the theory-driven equations and have added a new figure to the main text to contextualize the theory-driven results.

It is certainly reasonable to expect that site and season specific calibrations will do better than generic calibration equations. This is stated as a conclusion in the abstract, but its not clearly shown in the manuscript (except in Figure 5). The authors need to provide a table showing R2, NRMSE and bias for the SFS generated values along with those from a more generic calibration, such as that used by the US EPA, either Barkjohn 2021 or the updated EPA equation given here: https://cfpub.epa.gov/si/si_public_record_report.cfm?dirEntryId=353088&Lab=CEMM

This would be key to showing that site specific calibrations actually matter. The magnitude matters here. One could argue that improvements of a percent or so in NRMSE or 0.01 in the R2 are rather trivial.

We thank the reviewer for pointing us to add context to the significance of our results. We have added to the SI (Tables S4-S5) demonstrating the relevance of our model forms compared to applying the EPA correction from the peer-reviewed Barkjohn et al 2021. We have added the following details to the Results section.

> In Barkjohn et al 2021a, the large sample size of PA-II across the continental United States was used to derive a similar calibration regression. In SI Tables S4-S5 we compare the NRMSE and MBE for our best CF1 model forms from the SFS procedure (up to 3 parameters), theory-driven CF1 model, and Barkjohn et al 2021 model output. We have found from our seasonally-balanced test dataset that our models perform moderately

better (ΔNRMSE of about 5% across sites) than the EPA model, which is perhaps intuitive given the differences in PM composition and concentrations in India relative to the US. Furthermore, our site-specific models' MBEs are close to 0 μg/m3, while the Barkjohn et al 2021 model systemically underestimates mass concentrations, with an MBE as high as 22 μg/m3 in Delhi, compared to an MBE of -0.7 μg/m3 using the Delhi site-specific model or 3.25 μg/m3 using the Hamirpur model on the Delhi test dataset. Overall, while the site-specific models we develop here clearly outperform the model of Barkjohn et al (2021a) for these three Indian sites, it is nonetheless striking that this US-developed calibration still performs quite well at these three Indian sites.

In addition, I don't understand how the authors went from the more complex calibration relationships shown in Table 2, to the much simpler relationships shown by equations 2, 3 and 4.

We agree that more details should be shared to demonstrate our model selection methods using more transparently the SFS procedure. We have added text and citations more clearly describing the procedure. We have also added a figure to the SI (the new SI Fig. S16), showing the impact of adding model parameters on adjusted $R^2$ – which we have already shown to be marginal after adding a second parameter.

To iterate across all possible arrangements of predictors - including additive terms, interaction terms, as well as polynomial terms up to order 3 – we implemented Sequential Feature Selection (SFS) using the Python package scikit-learn 0.24.2. SFS uses a "greedy" approach to converge on the best-performing model for a user-defined number of parameter (Raschka and Mirjalili, 2019; James et al., 2013; Ferri et al., 1994). For example, if a user wanted a 2-parameter model from a set of 10 features, SFS would iteratively compare 90 models, the set of all possible 2-parameter feature permutations, using a robust regression metric (such as adjusted $R^2$ or Bayesian Information Criterion [BIC]). In our approach, we first use SFS to define the best-performing n-parameter model starting with all possible parameters (n=34). We then compare adjusted $R^2$ across best-performing n-parameter models to measure the impact of model complexity. If increasing parameters results in only marginal improvements ($\Delta R^2 \approx 0.01$), then it is unnecessary to use those additional features. The overall most robust model, therefore, reflects both the best possible selection of features as well as feature parsimony.

Temperature and dewpoint terms only imparted marginal improvements to calibration models ($\Delta R^2 \approx 0.01$, see SI Fig. S16), and it is not determinable if the models are deriving a spurious correlation or detecting underlying aerosol or instrument properties.

Other comments:

Line 76:   need to provide equations for "Alt" corrections in SI.

The ALT data is now common in literature as well as directly available from PurpleAir, therefore we direct readers to Wallace et al 2021 and Wallace et al 2022. SI Fig S1, panel b illustrates the

relationship between the ALT values and CF1 data. We have added detail to the methods section as well.

> Briefly, the ALT method adds all the particle counts from bins less than 2.5 μm, and calculates the particle volume concentration assuming spherical particles. Particle volume concentration is then multiplied by unit density (1 g/cm$^3$) to estimate PM$_{2.5}$ mass concentration.

77: what does three refer to?

We have clarified be replacing "three" with "CF1, ATM, and ALT."

83, 90: grammar issues.

We have fixed these grammatical errors.

150: What are "unreasonably large"?   What is "block average"?  Same as hourly average?

"Unreasonably large" is defined in Line 153 as the operational range (5- 500 μg/m$^3$)of the instrument as described in the cited literature.

We have reviewed the entirety of the text and replaced "block average" with "hourly average."

155: Not sure what you mean by statistically paired.

Statistically paired refers to independent datasets for each observation that can be considered "coupled" or matched because they are from the same location and are therefore expected to draw from the same underlying distribution. We have rephrased for clarity.

> Analyses of PurpleAir data typically report the percent error between channels A and B for a given unit to remove imprecise points, treating them as joint measurements and all other nodes as independent…

164: I don't think BME 280 is defined anywhere.

BME280 is not an initialism, it simply refers to the model's name. We have ensured that is properly contextualized here.

> The Adafruit model BME280…

177:  Not just RH, many other factors.

We have added another sentence explaining the roles of other key factors such as particle size distribution and index of refraction – although RH strongly moderates these factors, especially within the context of LCS observations.

> "…calibration procedures attempt to account for bias due to RH, index of refraction, and mischaracterizing the particle size distribution…"

211: Grammar.

We have fixed this grammatical error.

227: Suggest a citation for defs of these std relationships. (MBE, NMBE, etc)

We have used the citation Simon et al. 2012, and James et al. 2013 to clarify these points.

235: This is not a bias.

We understand this term may be confusing and instead have referred to it as mean difference.

> "… a mean difference from the regulatory network average of…"

284: Suggest this ref for quantitative analysis of dust with Pas

https://doi.org/10.5194/amt-16-1311-2023

Thank you for sharing, we have added it to our discussion on the impact of dust in the results and conclusion sections.

292: Table 2 does not summarize the procedure, but rather results.

We have clarified this in caption by replacing "procedure" with "results."

294: This does not seem to be true for Bangalore, PM x temp is the most relevant, right?

We have clarified that all sites selected a term which included a PM parameter.

> " The form of the most robust Bangalore model is different from the North India sites with an interaction term between temperature and ALT…"

295: I assume this refers to RH SQUARED, right?

Thanks, we have updated our formatting to ensure that the superscripts are properly formatted.

323-325: Not sure what this refers to.

We have added detail to the methods section to document the different data channels more precisely and clearly.

PurpleAir reports mass concentrations from PA-IIs in three forms, referred to as CF1, ATM, and ALT. CF1 ("Correction Factor 1") is the "uncorrected" data from the Plantower. The CF1 data has been demonstrated to strongly correlate with collocated integrating nephelometer data (Ouimette et al., 2021). ATM or "Atmospheric Corrected" data uses a piece-wise function to attempt to account for overestimation. SI Figure S1 illustrates this function across the full dynamic range for data collected in Delhi. Between 0 - 25 µg/m3, the CF1 and ATM data are 1:1, between 25 - 40 µg/m3 the ATM to CF1 ratio transitions from 1:1 to approximately 0.7:1, and at greater than 40 µg/m3 the ATM to CF1 ratio is stable at 0.7:1. Although it is reasonable to hypothesize the ATM data may better represent exposure ambient PM2.5 than the CF1 data, there is no transparent reasoning in the user manual for this design choice (Wallace et al., 2021; Zhou and Zheng, 2016). Finally, the ALT data represents a reconstruction of the PM2.5 data from the particle number data reported by the Plantower....Wallace et al. (2021) and Wallace et al. (2020) used this data to develop calibration relationships, reporting the ALT data as more transparent than using the CF1 or ATM data. However, the particle number data is known not to reflect the actual ambient size distribution since the Plantower PMS5003 is not a particle sizing instrument, but rather reflects a modeled size distribution using assumptions for relationships between size bins that is not always accurate for atmospheric conditions (Ouimette et al., 2021; Hagan and Kroll, 2020; He et al., 2020; Kuula et al., 2020). SI Figure S1 shows the ALT to CF1 ratio is approximately 0.15:1.

350 and equations 2,3 and 4:

After describing the details of a multi-linear SFS model, and showing in Table 2 how various permutations of the parameters give the best fits, I don't understand how you arrive at these much simpler relationships. These look like very standard PA calibration equations that have been developed by others.

As described in detail in our response to major comments, the SFS approach is "greedy" – it considers all possible permutations for a given number of features. Perhaps unsurprisingly given the underlying instrument and aerosol properties, when we limited models to only 2 parameters, we arrived at the same model formulation in Delhi and Hamirpur as other studies. Using more than 2 parameters via SFS offered only marginal improvements (see the new SI Figure S16). Therefore, although we started with many features and possible model forms, we converged at a form common to the literature. The broad continuity of the calibration form across geographies is valuable information for the community in India, since $PM_{2.5}$ concentrations and speciation are much different than in North America, Europe, and Australia.